# IAEA Contribution to Nanosized Targeted Radiopharmaceuticals for Drug Delivery

**DOI:** 10.3390/pharmaceutics14051060

**Published:** 2022-05-15

**Authors:** Amir R. Jalilian, Blanca Ocampo-García, Wanvimol Pasanphan, Tamer M. Sakr, Laura Melendez-Alafort, Mariano Grasselli, Ademar B. Lugao, Hassan Yousefnia, Clelia Dispenza, Siti Mohd Janib, Irfan U. Khan, Michał Maurin, Piotr Ulański, Say Chye Joachim Loo, Agnes Safrany, Joao A. Osso, Adriano Duatti, Kattesh V. Katti

**Affiliations:** 1Department of Nuclear Sciences and Applications, International Atomic Energy Agency (IAEA), 1400 Vienna, Austria; a.jalilian@iaea.org (A.R.J.); a.safrany@iaea.org (A.S.); j.a.osso-junior@iaea.org (J.A.O.J.); 2Departamento de Materiales Radiactivos, Instituto Nacional de Investigaciones Nucleares, Ocoyoacac 52750, Mexico; blanca.ocampo@inin.gob.mx; 3Department of Materials Science, Faculty of Science, Kasetsart University, Bangkok 10900, Thailand; fsciwvm@ku.ac.th; 4Radioactive Isotopes and Generators Department, Hot Labs Center, Egyptian Atomic Energy Authority (EAEA), Cairo 13759, Egypt; tamer_sakr78@yahoo.com; 5Istituto Oncologico Veneto IOV-IRCCS, Via Gattamelata 64, 35138 Padova, Italy; laura.melendezalafort@iov.veneto.it; 6Laboratorio de Materiales Biotecnológicos (LaMaBio), Dpto de Ciencia y Tecnología, Universidad Nacional de Quilmes, Bernal 1876, Argentina; mariano.grasselli@unq.edu.ar; 7Instituto de Pesquisas Energéticas e Nucleares, IPEN-CNEN, Av. Prof. Lineu Prestes, No. 2242, Cidade Universitária, São Paulo 05508-000, SP, Brazil; ablugao@gmail.com; 8Nuclear Science and Technology Research Institute (NSTRI), Tehran 11155-3486, Iran; hyousefnia@aeoi.org.ir; 9Dipartimento di Ingegneria, Università deli Studi di Palermo, 90133 Palermo, Italy; clelia.dispenza@unipa.it; 10Medical Technology Division, Malaysian Nuclear Agency, Kajang 43000, Malaysia; najila@nuclearmalaysia.gov.my; 11Cyclotron and Allied Radiopharmaceutics Division, Institute of Nuclear Medicine and Oncology (INMOL), New Campus Road, Lahore 54600, Pakistan; drirfankhan69@gmail.com; 12National Centre for Nuclear Research, Radioisotope Centre POLATOM, 05-400 Otwock, Poland; m.maurin@polatom.pl; 13Institute of Applied Radiation Chemistry, Lodz University of Technology, 90-924 Łódź, Poland; ulanskip@mitr.p.lodz.pl; 14School of Materials Science & Engineering (MSE), Spore Centre of Environmental Life Sciences Engineering (SCELSE), Lee Kong Chian School of Medicine, Harvard T.H. Chan School of Public Health, Nanyang Technological University (NTU), Singapore 639798, Singapore; joachimloo@ntu.edu.sg; 15Department of Chemical and Pharmaceutical Sciences, University of Ferrara, 44121 Ferrara, Italy; dta@unife.it; 16Department of Radiology, Institute of Green Nanotechnology, School of Medicine, University of Missouri, University of Missouri Research Reactor (MURR), Medical School, One Hospital Drive, Columbia, MO 65212, USA

**Keywords:** nanoparticles, metallic, non-metallic, polymeric, radiopharmaceuticals, radioisotopes, nanogels, theranostic, drug delivery, nanoradiopharmaceuticals

## Abstract

The rapidly growing interest in the application of nanoscience in the future design of radiopharmaceuticals and the development of nanosized radiopharmaceuticals in the late 2000′s, resulted in the creation of a Coordinated Research Project (CRP) by the International Atomic Energy Agency (IAEA) in 2014. This CRP entitled ‘**Nanosized delivery systems for radiopharmaceuticals’** involved a team of expert scientist from various member states. This team of scientists worked on a number of cutting-edge areas of nanoscience with a focus on developing well-defined, highly effective and site-specific delivery systems of radiopharmaceuticals. Specifically, focus areas of various teams of scientists comprised of the development of nanoparticles (NPs) based on metals, polymers, and gels, and their conjugation/encapsulation or decoration with various tumor avid ligands such as peptides, folates, and small molecule phytochemicals. The research and development efforts also comprised of developing optimum radiolabeling methods of various nano vectors using diagnostic and therapeutic radionuclides including Tc-99m, Ga-68, Lu-177 and Au-198. Concerted efforts of teams of scientists within this CRP has resulted in the development of various protocols and guidelines on delivery systems of nanoradiopharmaceuticals, training of numerous graduate students/post-doctoral fellows and publications in peer reviewed journals while establishing numerous productive scientific networks in various participating member states. Some of the innovative nanoconstructs were chosen for further preclinical applications—all aimed at ultimate clinical translation for treating human cancer patients. This review article summarizes outcomes of this major international scientific endeavor.

## 1. Introduction

The versatile surface chemistry of biocompatible nanomaterials has allowed the development of innovative nanodrug delivery systems for use in molecular imaging and therapy of various diseases. For example, the surface chemistry of gold nanoparticles (AuNPs) allow for efficient loading of DNA or RNA onto the surfaces via strong and biocompatible covalent interactions [1,2]. Tumor specific targeting abilities of cancer drug functionalized nanomedicine agents are proving to be highly effective in alleviating debilitating effects of adverse toxicity and drug resistance of standard FDA approved cancer therapy agents [3,4,5,6,7,8,9]. Several recent investigations have shown compelling scientific evidence that nanosizing of pharmaceuticals results in reducing onset of cancer drug resistance, and site specificity reduces systemic drug toxicity, offering significant improvement in therapeutic efficacy of current and future chemotherapeutic agents [6,7,8,9,10,11]. In sharp contrast to recent successes of nanosizing various pharmaceutical agents, currently used radiotherapeutic agents in nuclear medicine continue to pose medical challenges mainly due to the limited uptake of diagnostic/therapeutic radioprobes within tumor sites [12]. Problems associated with effective delivery of nuclear medicine radiotherapeutic drugs pose severe oncological challenges, especially when treating solid tumors (sarcomas, carcinomas, and lymphomas) which account for over 85% of all human cancers. Circumventing these problems is not easy because molecular and cellular biology of neoplastic cells alone fails to explain the non-uniform uptake of these agents in solid tumors. Repeated delivery of radiopharmaceuticals is not an option due to long lasting systemic radiotoxicity, creating major collateral adverse effects where cancer cells mutate, making them resistant to radiation therapy, chemotherapy and other treatments. Specifically, various beta-emitting therapeutic nuclear medicine agents have failed to deliver optimum therapeutic payloads at tumor sites. Therefore, the discovery of new classes of radiopharmaceutical drug delivery approaches that effectively penetrate extracellular compartments, consisting of vascular and interstitial valves within solid tumors, is of profound importance.

I-131 shows remarkably high tumor accumulation compared to any other radionuclide for diagnosis or therapy. I-131 is a widely accepted theranostic nuclear medicine agent extensively used to aid diagnosis and treatment especially for pre- and post-therapy scans in differentiated thyroid cancer patients [13]. This approach provides accurate surveillance of the disease—thus guiding clinicians for the treatment [13]. It is important to recognize that the high and specific uptake of diagnostic/therapeutic I-131 agent in tumors is a rarity because I-131 agents and a vast majority of nuclear medicine agents fail to localize in optimum doses in neo vascular lesions. This means that metastases of several different types of aggressive cancers cannot be detected accurately or treated through currently available nuclear medicine agents—thus resulting in the propagation of cancers to various other organs including—ultimately resulting in death in cancer patients’ population. Therefore, there is an overarching urgent need in the creation of new radiotherapeutic delivery modalities that offer: (i) effective delivery of radiotherapeutic probes with optimum payloads—specifically at tumor sites—with minimal/tolerable systemic toxicity, and (ii) therapeutically optimum tumor retention. Such innovative approaches would bring about a clinically measurable shift in the way cancers are diagnosed and treated. Nanotechnology has the potential to bring about this paradigm shift in the early detection and therapy of various forms of human cancers because radioactive NPs, of optimum sizes and conjugated to tumor specific targeting vectors, can be engineered to achieve effective penetration across tumor cell membranes. The disruptive and interdisciplinary nature of the field of nanotechnology is attracting the attention of a myriad of interdisciplinary professional teams of experts from nuclear medicine, materials sciences, physics, chemistry, tumor biology and oncologists engaged in testing the efficacy of nanosized radiopharmaceuticals in pursuits of achieving optimum diagnostic or therapeutic payloads of nuclear medicine agents. 

Radiolabeled tumor specific, peptide-conjugated nanopharmaceuticals can be engineered in sizes that would allow effective and site-specific accumulation of diagnostic or therapeutic probes in tumor cells/lesions through dual receptor specific endocytosis, as well as through passive enhanced permeation and retention (EPR) effects. In order to validate the hypothesis that nanosized radiopharmaceuticals provide the best means to achieve optimum accumulation and site, specifically of radioactive diagnostic/therapeutic probes in tumor sites, the IAEA launched an innovative Coordinated Research Project (CRP) entitled ‘Nanosized delivery systems for radiopharmaceuticals’ in 2014. The overarching objective of this CRP centered around the development of radiolabeled nanomedicine agents for diagnosis, therapy and theranostics of human cancers. As an international platform for peaceful applications of nuclear science and technology, the IAEA promotes and supports member states in various ways, including various international research projects, such as this CRP. The research, development, and collaborative activities of interdisciplinary scientists from 15 member countries participating in the CRP (Figure 1) focused on the development of new radiolabelled polymeric/hydrogel NPs, and also the radioactive Au-198-based nanoceuticals. This CRP is highly relevant for cancer diagnosis and treatment because targeted radiolabelled NPs would be ideal for homing in and penetrating tumor vasculature to provide optimum therapeutic/diagnostic payloads to solid tumors. Such an approach has the potential to minimize/stop metastases because once the primary tumors are destroyed, the tumor specific NPs would stop the recruitment of proliferating tumor cells into the bone marrow. This review is a culmination of innovative research findings of the international team of researchers who worked collaboratively under the auspices of IAEA’s CRP (Figure 1 and Figure 2). 

## 2. Radionuclides for Radiolabeling Various Nanoparticles 

Labeled NPs conjugated to specific biomolecules belong to an innovative class of diagnostic or theranostic radiopharmaceuticals. A nanoradiopharmaceutical is a multimeric compound comprises a nanoparticulate system, a radionuclide (including radioactive NPs) and a targeting molecule. Their potential applications are numerous, including targeted radioactive drug delivery systems for radiotherapy, as well as diagnostic imaging agents—all in a single nanoplatform.

To design a new nanoradiopharmaceutical, it is important to select the most appropriate radionuclide with an emission type (α, β^−^, β^+^ or γ) that will confer appropriate characteristics of diagnostic, therapeutic or theranostic payloads. The most important properties to consider when selecting a suitable radionuclide are the emission mode, half-life, chemical properties, ease of production and availability of target pre-irradiation materials. Table 1 summarizes various radionuclides used in the architecture of nanoradiopharmaceuticals discussed in this review. The radiolabelling can be carried out by mainly two methods: (i) through creation of covalent bonds between radionuclide(s) and targeting vectors, or (ii) through a bifunctional chelating agent approach such as DOTA, HYNIC [14]. Generally, radiolabeling strategies, using a bifunctional chelating agent, are preferred due to their higher in vivo stability—thus resulting in higher tumor specificity. 

^99m^Tc is a radionuclide, which has been most widely used to label tumor specific biomolecules, that presents tremendous potential for labelling NPs for developing SPECT imaging agents. Its gamma-emission has ideal properties for single photon emission computed tomography (SPECT) and would be ideal for development of diagnostic nanoradiopharmaceuticals. This radionuclide has been used extensively in nuclear medicine for the development of target-specific molecular imaging of alpha(v)beta(3) integrins, somatostatin receptors, gastrin releasing peptide receptors (GRPr), for sentinel node detection and a myriad of receptor positive tumors [15,16,17,18,19,20], as well as in imaging of cancers that overexpress Scavenger Receptor type B1 [21]. ^99m^Tc is also useful to follow the doxorubicin delivery from ^99m^Tc-DOX-loaded GA-Au NPs [22] and ^99m^Tc-Doxorubicin-Epigallocatechingallate functionalized gold nanoparticles [23]. In addition, ^99m^Tc can also be considered a theranostic radionuclide if its Auger electrons are emitted within the cell nucleus because, despite the low energy of its Auger electrons, they can be deposited in a very small area. Consequently, if Auger electrons are emitted close to the DNA, its biological effects might mimic that of α particles, causing inhibition of cell proliferation or even cell death [24,25]. In this review, we confine discussions of nanoradiopharmaceutical to those developed through the IAEA CRP, which includes ^99m^Tc, but also ^68^Ga and ^198^Au, and allied diagnostic and therapeutic radioisotopes (see Table 1).

Radionuclides including ^67^Cu, ^131^I, ^153^Sm, ^177^Lu, ^188^Re and ^198^Au that decay emitting a combination of β^−^ particles are suitable for therapy, and those that emit γ-rays are useful for SPECT imaging. ^177^Lu is one of the most widely used radionuclides for labeling radiopharmaceuticals due to its attractive decay properties. This radionuclide has a relatively long half-life (6.65 d) and emits a combination of β^−^ particles with Emax = 0.497 MeV (76%) and low-energy gamma photons Eγ = 113 keV (6.6%) and 208 keV (11%) [26]. Furthermore, ^177^Lu can form very stable complexes with several commercially available chelating agents including DOTA derivatives. Therefore, this radionuclide has been used to radiolabel several target-specific nanopharmaceuticals for therapeutic purposes [22,27,28], as well as for combinatorial oncotherapeutic drug delivery and targeted radiotherapy [29,30,31]. This approach has resulted in synergic effects between targeted radiotherapy and plasmonic photothermal therapy [32,33].

Nanoparticles intrinsically labeled using radionuclides such as ^64^Cu, ^72^As, ^111^In, ^153^Sm, ^177^Lu and ^198^Au, without employing chelators, have also been reported [34,35,36,37]. ^198^Au emits a beta energy (0.96 MeV) ideal to penetrate within the tumor tissue up to 11 mm, providing cross-fire effects which can destroy the tumor cells [38]. Therefore, several ^198^Au-gold nanoparticles (^198^AuNPs) were developed in the framework of this project (see Table 1). ^198^AuNPs stabilized with epigallocatechin gallate (EGCG) and mangiferin (MGF) phytochemicals are some examples of nanoradiopharmaceuticals successfully translated to the clinical practice [38]. An attractive feature is that tumor specific ^198^AuNPs can be architectured through green nanotechnology approaches by direct irradiation of natural gold foil or metal Au-197 (n, γ) for breast and prostate cancer therapy (see Section 5 for details) [38,39,40,41,42].

**Table 1 pharmaceutics-14-01060-t001:** Radioisotopes for labelling various nanoradiopharmaceuticals.

Radionuclide	Half-Life	Decay Mode (Energy, Intensity)	Applications	Type of Nanoparticle	Ref.
Ga-68	67.7 m	β^+^ (1.89 MeV, 88%)	PET imaging	^68^Ga-DOTA-BN-TMC-MNPs	[43]
Tc-99m	6 h	γ-ray (140 keV)	SPECT imaging	^99m^Tc-AuNPs ^99m^Tc-AuNP-Tyr^3^-Octreotide ^99m^Tc-DOX-AuNPs ^99m^Tc-AFCuONPs ^99m^Tc-SeNPs ^99m^Tc-VitC-SeNPs ^99m^Tc-MIONPs ^99m^Tc-PAMAM-Tyr^3^-Octreotide	[44,45] [17,46] [22,47] [48] [49] [50] [51] [17,46]
I-131	8 d	β^−^ (606 keV, 90%) γ-ray (364 keV, 81%)	Therapy and SPECT imaging	^131^I-doped Ag-PEG NPs	[52]
Sm-153	46.3 h	β^−^ (634 keV, 30%, 704 keV, 49% and 807 keV 20%) γ-ray (103 keV, 29%)	Therapy	^153^Sm-CSNPs-PEI-folate	[53]
Lu-177	6.6 d	β^−^ (496 keV, 80%) γ-ray (113 keV, 6% and 208 keV, 10%)	Theranostic	^177^Lu-DOTA-DN(PTX)-BN ^177^Lu-BN-PLGA(PTX) ^177^Lu-DOTA-HA-PLGA(MTX) ^177^Lu-DN(AuNP)-folate-BN ^177^Lu-DN(C19)-CXCR4	[29] [26] [54] [55] [31,56]
Au-198	2.7 d	β^−^ (961 keV, 99%) γ-ray (411 keV, 96%)	Therapy	^198^AuNPs ^198^AuNPs-BSA MGF-^198^AuNPs AX-encapsulated ^198^AuNPs	[57] [58] [38,42] [59]
Au-199	3.1 d	β^−^ (243 keV, 22% and 293 keV, 72%) γ-ray (158 keV, 40%)	Therapy	^199^AuNPs	[57]

## 3. Design and Development of Nanoparticles

Due to unprecedented diagnostic/therapeutic multiplexing capabilities, functionalized NPs have the potential to play a key role in delivering various radionuclides for nuclear imaging and therapy. By tailoring the properties of NPs, such as chemical composition, size, shape and surface properties, including tumor site-specific targeting molecules, NPs can be rationally designed down to the single-molecular level to achieve desirable properties. Such properties include nanoparticulate stability, optimal pharmacokinetics, tissue penetration, cell internalization, and targeting efficacy. Inorganic metals and polymers have served as important building blocks for constructing NPs as drug delivery systems. In addition, hybrid NPs comprising both metal and polymeric components have also been considered for nanosized delivery because of the combined benefits of co-delivery of both diagnostic and therapeutic agents. Figure 3 illustrates NPs developed from inorganic metals and polymers using chemical-based and radiation-based synthesis under this IAEA CRP. By tailoring molecular structures using suitable synthetically designed methods, nanoparticles with a myriad of nanostructures including inorganic NPs, hybrid NPs, dendrimer/multibranched structures, nanogels from both synthetic and biopolymers, amphiphilic or core-shell NPs from graft copolymer and functional polymers, can be developed for use in radiopharmaceuticals delivery.

### Chemical-Based Synthesis of Nanoparticles—General Aspects of Production

Formation of chemical-induced metallic NPs are usually based on conventional chemical reduction and phytochemical-induced reduction mechanisms. For polymeric NPs, polymerization, copolymerization, graft copolymerization, crosslinking and chemical functionalization are used for tailoring polymeric structures for the creation of NPs under appropriate conditions. Selected NPs from chemical-based synthesis are listed in Table 2, along with their size and summary of synthesis.

Inorganic NPs include metallic NPs, metal oxide nanoparticles, magnetic NPs, and quantum dots [63,64,65,66]. Inorganic NPs can be chemically synthesized using several different techniques such as co-precipitation, hydrothermal, combustion, sol-gel, chemical reduction, chemical vapor and microwave-assisted methods [67,68,69,70,71]. In this review, we present inorganic NPs that are synthesized using reducing and stabilizing agents where the reducing agent is responsible for the reduction of the metal from high oxidation states (+3, +2 or +1) to its metallic form (zero oxidation state), while the stabilizing agent is responsible for surface stabilization of metal nano materials [22,42,44,45,47,49,50,52]. Various NPs including AuNPs, selenium nanoparticles (SeNP), iron nanoparticles (FeNP), copper nanoparticles (CuNP) and silver nanoparticles (AgNP) were evaluated for their effective delivery of different radionuclides such as tecnetium-99m (Tc-99m), iodine-131 (I-131) and gold-198 (Au-198) to tumor sites for tumor theranostics [42,44,45,47,48,49,50,51,52,72]. 

Multimeric systems based on polymeric NPs have also been designed, synthesized, characterized and evaluated for targeted combinatorial radiotherapy and chemotherapy. These type of nanoradiopharmaceuticals can deliver therapeutic radiation doses including specific drugs to tumor sites, thus reducing collateral damage to healthy tissue. Under this CRP, several radiolabelled nanoprobes have been studied as promising radiopharmaceuticals for targeted chemo/radiotheranostic applications. In this context, specific nanoradiopharmaceuticals based on dendrimers and PLGA have been conjugated to targeting biomolecules such as CXCR4L [31], bombesin [29,30], RGD peptide [62] and folic acid [61] to specifically target tumors which over-express tumor specific receptors. Hyaluronic acid was conjugated to PAMAM dendrimers to target CD44 receptors on activated macrophages in rheumatoid arthritis [54]. DOTA (1,4,7,10-tetraazacyclododecane-1,4,7,10 tetraacetic acid) has been used as a bifunctional chelator for a Lu-177 beta-emitter.

## 4. Radiation-Based Synthesis of Nanoparticles 

A radiation-induced chemical reactions approach is one of the effective and green processes well-suited for the development of biomedical nanomaterials. This approach presents excellent opportunities for tailoring varieties of nanostructured morphologies of inorganic materials and polymers. This technique is versatile due to its simplicity and fast reaction kinetics/operations, and is catalyst free. Radiation-induced chemical reactions enjoy additional unique advantages as they employ powerful radiolysis for controlling chemical reactions, which also simultaneously provides sterilization processes. Radiation-based synthesis for NPs construction can be carried out through degradation, crosslinking, graft copolymerization, and also radiation-induced reduction of metallic ions. This technique can be used for generating (i) radiation-crosslinked nanogels from synthetic/biopolymers [73,74,75,76,77,78,79,80,81,82,83,84], (ii) conjugated NPs from size-controlled polymer templates through radiation [85,86], (iii) NPs from self-assembly of radiation-induced grafted copolymers [87,88], and (iv) AuNPs-hybrid NPs from radiation-induced reduction [89,90,91,92,93,94]. These are summarized in Table 3. For example, the crosslinked poly(acrylic acid) (PAA) nanogels prepared from pulse radiolysis has been studied through careful control of particle sizes by absorbed dose and dose rate [73,74,75,95]. The interpolymer-complex (IPC) nanogels composed of two polymer components is also a promising multifunctional-designed nanostructure [76]. 

Radiation-crosslinking techniques have also been applied for synthesis of IPC nanogels from polyethylene oxide-PAA (PEO-PAA) systems, with pH- and temperature-responsive functions [76] as smart nanocarrier approaches. With the carboxylic groups (−COOH) of PAA and responsive functions, controlled delivery of radiopharmaceuticals and tumor target specific peptide conjugates are possible. Grafting polyvinyl pyrrolidone (PVP)-based nanogels and conjugating with functional molecules, such as acrylic acid, folic acid, amino-terminated groups, (3-N-aminopropyl) methacrylamide hydrochloride, insulin, and monoclonal antibody, was applied to develop functional and selective nanocarriers [82,83,84,96,97,98,99,100]. In addition to the synthetic polymeric nanogels, radiation-induced crosslinking of protein-based NPs was studied under controlled and varying irradiation conditions. An albumin nanogel was successfully established for applications as therapeutic drug carriers [77,78,79,80,81]. 

Particle size of NPs can be controlled using polymer chain templates. Radiation-controlled polymer-chain templates for controlling particle size of deoxycholic acid functionalized chitosan NPs as a drug nanocarrier was demonstrated [85,86]. Radiation-induced graft copolymerization of stearyl methacrylate (pSMA) or poly(poly(ethylene glycol) monomethacrylate) (pPEGMA), as hydrophobic/hydrophilic polymer brushes on chitosan, have been reported [87,88]. In this way, building blocks of grafted copolymers showed their ability to self-assemble into confined amphiphilic core-shell NPs for antibiotic drug encapsulation and antioxidant molecule functionalization. With plenty of functional hydroxyl (−OH) and amino (−NH_2_) functional groups on chitosan, chitosan-based NPs exhibited availability for conjugating with radionuclide-chelating-molecules and targeted peptide biomolecules [91]. Nano-templates of water-soluble biopolymer polysaccharide and polypeptide from radiation-controlled processes were applied for one-pot synthesis of AuNPs [89,90,91,101]. Recently ready-to-use templates for radioactive ^198^AuNPs, using biopolymers incorporating tumour-targeted peptides, have been successfully developed for use as nanomedicine agents in molecular imaging and therapy [91]. It has been proven that core-shell AuNPs can be efficiently synthesized in albumin protein solution under irradiation [92,93,95].

Radiation-based synthesis of NPs allow systematic variations in functions and properties of NPs for nanoradiopharmaceutical applications. Under the controlled-synthesis process (e.g., absorbed dose, dose rate, polymer characteristics, polymer-size template, and grafted monomer), the particles sizes of the confined nanostructures can be controlled. The particle sizes of metallic/hybrid NPs and polymeric NPs generated were found in the ranges of 5–100 nm and 20–300 nm, respectively. With their specific characteristics and functionalities, these NPs provide prospects for drug encapsulation and, as radioisotope carriers, for tumour-targeting through theranostic modes.

**Table 3 pharmaceutics-14-01060-t003:** List of NPs developed by radiation-based synthesis.

Categories of NPs	Name of NPs	Size (nm)	Information of NPs	Ref.
Polymeric	PAA nanogels	30–200 ^a^	Nanogels of poly(acrylic acid) synthesized by preparative pulse radiolysis and decorated with bombesin/DOTA, tested for radioisotope binding and subjected to preliminary tests on animal model (mice).	[73,74,75]
	PAA-PEO IPC nanogels	~100–240 ^a^	Poly(acrylic acid)-poly(ethylene oxide) interpolymer (IPC) complex nanogels exhibiting pH- and temperature-responsive functions and possible for drug-controlled release.	[76]
	PVP-g-AA-Folate PVP-g-AA-Doxo PVP-g-AA-SiRNA	20–60 ^a^	Polyvinyl pyrrolidone based nanogels with acrylic acid grafts: (i) decorated with folic acid (FA) for preferential uptake by cells that overexpress folate receptors; (ii) conjugated, via a redox-cleavable linker, to either doxorubicin or a silencing RNA.	[82,83,84]
	PVP-g-AA-AntiMIR	~50 ^a^	Polyvinyl pyrrolidone-based nanogels with acrylic acid grafts conjugated to the amino-terminated AntimiR-31, that targets MiR-31, a microRNA overexpressed by primary and metastatic tissue colon cancer cells (CCR) and target of the E2F2 gene that plays a crucial role in the control of CCR progression.	[100]
	PVP-g-AA-Insulinl	~80 ^a^	Polyvinyl pyrrolidone-based nanogels with acrylic acid grafts, conjugated to insulin to be intranasally delivered, bypass the blood–brain barrier and target the brain.	[96,97]
	PVP-g-APMAM	80–300 ^a^	Polyvinyl pyrrolidone-based nanogels with (3-N-aminopropyl)methacrylamide hydrochloride grafts, conjugated to a monoclonal antibody which recognizes the αvβ3 integrin, a receptor important in tumor angiogenesis and metastasis.	[98,99]
	Protein NPs	20–40 ^a^	Albumin NPs prepared by a novel radiation-induced crosslinking method. Albumin preserve their original drug-carrier properties.	[77,78,79,80,81]
	CS-DC NPs	50–100 ^b^ 30–50 ^b^	Deoxycholate conjugated chitosan NPs containing hydrophobic core for water-insoluble drug encapsulation. NPs remaining -OH and -NH_2_ groups for further conjugating with chelating/peptide molecules.	[85,86,91]
	pPEGMA-CS-DC NPs	70–130 ^b^	Poly(ethylene glycol) monomethacrylate grafted chitosan deoxycholate amphiphilic NPs containing hydrophobic core for water-insoluble drug encapsulation (e.g., berberine). NPs remaining -OH and -NH_2_ groups for further conjugating with chelating/peptide molecules.	[87,91]
	pSMA-CS NPs	50–140 ^b,c^	Poly(stearyl methacylate) grafted chitosan NPs providing -OH and -NH_2_ groups for further conjugating with chelating/peptide molecules. NPs having hydrophobic core for water-insoluble drug encapsulation.	[87,91,102]
	pSMA-CS-PPD NPs	50–100 ^b^	Piperidine conjugate poly(stearylate)-grafted chitosan NPs exhibiting antioxidant function and remaining -OH and -NH_2_ groups for further conjugating with chelating/peptide molecules.	[88,91]
	WSCS nanocolloids	49 ± 2.15 ^a^ 10–50 ^b^	Water-soluble chitosan nanocolloids exhibiting antioxidant activities and reducing power. NPs remaining -OH and -NH_2_ groups for conjugating with chelating/peptide molecules. NPs enable use as biopolymer-template synthesis of Au-197 and Au-198 analogue.	[89,91]
	SF nanocolloids	~40 ^b^	Silk fibroin (SF) nanocolloids exhibit antioxidant activities and reducing power. NPs enable use as biopolymer-template synthesis of Au-197 and Au-198 analogue.	[101]
	WSCS-DOTA-BBN NPs	86 ± 2.03 ^a^	Water-soluble chitosan conjugated DOTA NPs chelator and BBN peptide enable use as one-pot synthesis of targeted AuNPs.	[91]
Inorganic/hybrid	AuNPs-CS	5–80 ^b^	Protocol synthesis of AuNPs capped with chitosan enable use for one-pot synthesis of ^198^Au nanoradiotherapeutics.	[89,90]
	AuNPs-WSCS	5–25 ^b^	Protocol synthesis of AuNPs capped with water-soluble chitosan enable use for one-pot synthesis of ^198^Au nanoradiotherapeutics.	[90]
	AuNPs-WSCS-DOTA-BBN	62 ± 21 ^a^ 20 ± 7 ^b^	AuNPs capped with water-soluble chitosan conjugated DOTA chelator and BBN peptide acting as targeted therapeutic agent for prostate cancer cells (i.e., PC-3, LNCaP). Protocol for a one-pot synthesis of the targeted ^198^Au nanoradiotherapeutics.	[91]
	AuNPs-WSCS-GA-DOTA-BBN	40–60 ^a^	BBN peptide conjugated water-soluble chitosan gallate as a new nanopharmaceutical architechture for the rapid one-pot synthesis of prostate tumor targeted AuNPs	[103]
	bioHNPs	77 ± 7 ^a^	AuNPs coated with human albumin multilayer and further decorated with DOTA chelator and BBN peptide acting as targeted therapeutic agent for prostate cancer cells (i.e., PC-3).	[92,93,94,95]

^a^ Hydrodynamic size, ^b^ Size determined by TEM, ^c^ Size determined by AFM.

Our work on mangiferin-conjugated radioactive gold nanoparticles of Au-198 (MGF-^198^AuNPs) represents the most advanced research development because (i) we have optimized the production of Au-198 nanoparticles; (ii) performed extensive in vitro investigations on their stability and also investigated biodistribution in normal mice, and finally (iii) we have also evaluated tumor retention as well as therapeutic efficacy of MGF-^198^AuNPs in tumor bearing mice. Therefore, we have presented full details of preclinical/clinical investigations of this singular nanomedicine agent, as a representative candidate, in this review article.

## 5. Production, Characterization, and Pre-Clinical/Clinical Investigations of Radioactive MGF-^198^AuNPs

### 5.1. Production, and Characterization of Radioactive MGF-^198^AuNPs

Gold presents several isotopes including radioactive isotopes—Au-198 and Au-199—for medical applications. Au-198 with 2.70-day half-life emits both β (961 keV) and γ radiation (411 keV γ (96%)). Its half-life and beta energy emission allows efficient destruction of tumor cells/tumor tissue, as the β penetration range is ideal (up to 4 mm in tissue or up to 1100 cell diameters) and 2.7 d is sufficiently long to provide crossfire effects to destroy tumor cells/tissue, while short enough to minimize radiation exposure to neighbouring non-target tissues. A distinct advantage of radioactive gold NPs is that there is no need to incorporate the isotope into every tumor cell to have a therapeutic effect because the path length of the emitted radiation is sufficient to allow effective therapy following uptake into a subpopulation of tumor cells. This makes NPs of Au-198 ideal therapeutic candidates for the effective delivery of therapeutic doses of beta emitting NPs selectively to tumor tissue and tumor cells. Au-198 allows monitoring therapeutic response because its γ radiation allows scintigraphic imaging of various normal and tumor tissue. Scintigraphy approaches allow evaluation of tumor retention characteristics of AuNPs, labelled with small tumor specific biomolecules or conjugated to polymeric NPs (including hydrogels)—by determining the bio-distribution of radioactive gold nanoparticles post-administration of ^198^AuNPs. In order to understand the retention and pharmacokinetics of MGF functionalized gold nanoparticles (MGF-AuNPs) within prostate tumors, we have recently synthesized and fully characterized the corresponding radioactive analogue of MGF-^198^AuNPs by labelling MGF with ^198^AuNPs (Figure 4). Mangiferin displays tumor cell avidity with high selectivity toward receptors over-expressed by a host of tumors including prostate, breast, colorectal and pancreatic tumors [38,104,105]. γ emission characteristics of MGF-^198^AuNPs have been highly useful to estimate its retention within prostate and various other tumors and non-target organs. For the synthesis of radioactive MGF-^198^AuNPs, we first had to optimize the production of ^198^Au at the University of Missouri Research Reactor (MURR) facility in high radiochemical purity. The procedure optimized for the synthesis of Au-198 isotope is discussed in the following sections.

The production protocol irradiation of natural abundance high purity gold foil to produce ^198^Au according to the following nuclear equation Au-197(n,γ) Au-198 [38]. A neutron flux of 8 × 10^13^ n/cm^2^/s at MURR was used. The irradiation times and mass of gold foil were optimized depending on the activity of Au-198 isotope needed; typically, irradiation times were from 6 to 40 h. After each irradiation, the radioactive foil was dissolved in about 400 µL of aqua regia, and subsequently the mixture was heated to produce a dry residue. Following the addition of two 400-µL aliquots of 0.05 M HCl, the mixtures were heated to drive off the nitric acid. At this stage, the radioactive Au-198 was dissolved in optimum volumes of water to produce final solutions of ^198^Au for subsequent applications in the production of MGF-^198^AuNPs. 

### 5.2. Preparation of ‘Premix’

The radioactive gold (^198^Au) solution, as prepared above, was mixed with Na_2_AuCl_4_ to form radioactive gold precursor (Premix) to produce total mass of radioactive ^198^Au and non-radioactive Au of 0.66 mg, and a desired specific activity, according to the required activity in the final solutions of NPs [38]. For example, in our therapeutic investigations, the mass of ^198^Au was 0.082 mg (this mass was chosen because it had the required activity of 481 MBq). We therefore mixed 0.082 mg of ^198^Au with gold salt that had a mass of gold equal to 0.578 mg, in order to make the total mass of gold in the solution equal to 0.66 mg. All these mixtures were produced in a 2 mL solution. If lower activities were needed, then accordingly, the mass of ^198^Au was decreased while the mass of gold within gold salt increased, to maintain the total mass of gold at 0.66 mg [38].

### 5.3. Production of Radioactive MGF-^198^AuNPs

It is important to note that well-optimized procedures for the synthesis of the non-radioactive MGF-AuNPs, when applied toward the synthesis of the corresponding radioactive MGF-^198^AuNPs, did not work [105]. This is not surprising because the kinetics of chemical reactions at macroscopic levels often differ considerably from those at tracer levels. Therefore, we have developed a slightly modified procedure wherein the mass of MGF (1.4 mg for non-radioactive MGF-AuNPs preparation) was increased to 1.55–1.6 mg for the preparation of radioactive MGF-^198^AuNPs (Figure 4). We have shown that 1.55–1.6 mg of MGF produces radioactive MGF-^198^AuNPs in high radio-chemical purity with excellent reproducibility from batch to batch [38]. The production protocols involved addition of 1.55–1.6 mg of MGF to 2 mL of milli-Q water. The reaction mixtures were stirred at room temperature for 10 min, and subsequently heated under stirring at 99 °C. To this mixture of MGF, we added the radioactive gold Premix solution (^198^Au + Na_2_AuCl_4_), which has the desired activity (14.43–481 MBq). This addition instantaneously changed the color of the reaction mixture from pale yellow to burgundy red, indicating the formation of radioactive gold nanoparticles. Such color transformations in gold nanoparticle synthesis signify the appearance of surface plasmon resonances, which are characteristic photon absorptions of surface atoms of nanosized noble metal particles, including gold. For completion of this reaction, stirring was continued for an additional 1 h at room temperature.

### 5.4. Characterization

Radioactive MGF-^198^AuNPs were characterized by measuring the surface plasmon resonance (SPR) wavelength (λmax) using UV-Vis spectroscopy, which showed a λmax peak at 545 nm [38,105]. Thin-layer chromatography (TLC) measurements were used to determine radioactive yields of MGF-^198^AuNPs. Free ^198^Au solution moves to the solvent front in TLC, whereas ^198^Au-nanoparticles remain at the origin. The procedure was performed by adding 1 µL of nanoparticles solution to the origin of cellulose TLC plate. Accurate yields of radioactive gold (over 95%) were measured using a radio TLC (Bio-Scan company).

After the successful production of MGF-^198^AuNPs, the next logical step was to understand its biodistribution in normal mice. Accordingly, biodistribution of MGF-^198^AuNPs in normal mice was evaluated at various time points, as shown in Figure 5 [38]. It is clear from this biodistribution profile that (1) this nanomedicine agent has optimum in vivo stability with efficient clearance from blood and various non-target organs, and (2) MGF-^198^AuNPs clear through the hepatobiliary system, as indicated by its liver uptake (Figure 5). These biodistribution investigations provide convincing rationale of the utility of Au-198mγ, thus allows scintigraphic imaging for the accurate assessment of the of functionalized gold nanoparticles. Considering the availability of a vast repertoire of functionalized gold nanoparticles for potential applications in nanomedicine, this investigation highlights the need to produce the corresponding Au-198 labelled analogues, to enable scintigraphic imaging investigations for gaining invaluable insights on the fate of gold nanoparticles in vivo.

We further probed into the cancer cell receptor specific features of MGF-AuNPs because the pharmacomotif of MGF comprised of a glucose moiety and polyphenol (Figure 4) has strong avidity and specificity toward Laminin receptor proteins, which are signatures of various tumor cells.

### 5.5. Laminin 67 Receptor-Mediated Cellular Internalization of MGF-AuNPs

To study the internalisation of MGF-AuNPs into tumor cells, prostate tumor (PC-3) cells derived from human prostate tumors were incubated with MGF-AuNPs and subsequently characterized. Both transmission electron microscopy and dark field microscopy indicated that these NPs are effectively internalized into tumor cells. Further, extensive neutron activation analysis (NAA) results, to quantify gold content in PC-3 cells, as well as receptor blocking experiments, have established that the endocytosis of MGF-AuNPs within PC-3 cells are mediated through Lam 67 receptors [38,105]. These experiments show the selectivity of MGF-AuNPs for prostate cancer cells, a critical aspect to target primary disease without secondary morbidity. As shown in Figure 6, MGF-AuNPs are endocytosed into PC-3 cells showing localization in vacuoles, as well as in the cytoplasm without disturbing the nucleus. MGF-AuNP, which are homed into tumor cells, appear to remain intact, suggesting high avidity of MGF-AuNP toward laminin receptors over expressed by tumor cells [105]. The dark field microscopic images, as shown in Figure 6, corroborate TEM images—thus confirming endocytosis of MGF-AuNPs in tumor cells [105]. These results demonstrating the selectivity of MGF-AuNPs for prostate cancer cells, prompted a detailed pre-clinical in vivo tumor retention and therapeutic efficacy investigation of MGF-^198^AuNPs.

### 5.6. Pre-Clinical In Vivo Tumor Retention and Therapeutic Efficacy Investigations of MGF-AuNPs

Biodistribution of radioactive gold nanoparticles, post-administration of ^198^AuNPs, as synthesized above, was undertaken to evaluate the retention of these NPs in prostate tumors. We have performed in vivo studies of intratumoral administration of MGF-^198^AuNPs (148 kBq/30µL for each tumor) using SCID mice (n = 5) bearing human prostate cancer xenografts. The biodistribution and tumor retention characteristics are shown in Figure 7.

The biodistribution results have confirmed that the percentage of injected dose within tumors (%ID), especially within prostate tumors at various time points, was 80.98 ± 13.39% at 30 min, increasing to 86.68 ± 3.58% at 2 h, and remained at 79.82 ± 10.55% at 24 h. Overall, there was minimal/no leakage of MGF-^198^AuNPs in the liver. The (%ID) in liver was 10.65 ± 8.31% at 24 h, indicating that a small number of NPs were cleared by the reticuloendothelial system (RES) to the liver. In addition, there was very low leakage of injected dose into stomach and feces, 0.10 ± 0.16% of injected dose in stomach at 30 min decreasing to 0.02 ± 0.02% at 24 h, and 0.0% of injected dose in feces at 30 min increasing to 2.20 ± 4.51% at 24 h. These results show that the main route of clearance is via the digestive system through the feces. In contrast, there was no noticeable leakage into blood and lung and other organs. Therefore, these results from intratumoral investigations have demonstrated that MGF-^198^AuNPs have excellent ability to be retained within the tumor with very minimum leakage to non-target organs. The high tumor retention with concomitant limited leakage of MGF-^198^AuNPs are two key factors to minimize/eliminate systemic toxicity, and thus provide compelling scientific rationale for the future utility of this nanoradiopharmaceutical in tumor therapy.

### 5.7. Therapeutic Efficacy of MGF-^198^AuNPs Nanoradiotherapeutic Agent

Mice with severe combined immunodeficiency (SCIDs) bearing a flank model of human prostate cancer, derived from a subcutaneous implant of 10 million PC-3 cells, were used for therapeutic efficacy and pharmacokinetic studies of MGF-^198^AuNPs. Therapeutic efficacy data from the single-dose radiotherapy through intratumoral administration of MGF-^198^AuNPs (5 MBq) in PC-3 bearing SCID mice are shown in Figure 8. The end-of-study biodistribution on day 42 showed that 64.7. ± 6.5 %ID (mean ± sem; n = 5) remained in the residual tumor, while 13.8. ± 4.2 %ID was noted for carcass and 1.5. ± 0.8 %ID for the liver. Retention in other tissues was negligible, with radioactivity near background levels for blood, heart, lung, spleen, intestines, stomach, bone, brain and skeletal muscle. There was a significant reduction in tumor volume (87%), four weeks after a single intratumoral dose of MGF-^198^AuNPs with minimal uptake in non-target organs (Figure 8). The therapeutic efficacy demonstrated for MGF-^198^AuNPs confirm its ability to inhibit tumor growth, because tumors harvested from the treatment groups consisted largely of necrotic tissue, indicating extensive tumor cell death. The tolerability of MGF-^198^AuNPs in vivo has been established by monitoring the body weight and blood parameters in a SCID mice study, in both treated and control groups of animals. The treatment group showed only transient weight loss with recovery to normal weight without any early terminations. White and red blood cell, platelet, and lymphocytes levels within the treatment group resembled those of the normal mice without tumors. The overall health status and blood measures of the MGF-^198^AuNP-treated animals indicate that the radiotherapy was not only effective, but also well tolerated. These findings support the effectiveness of intratumoral delivery of the nanoradiotherapeutic agent MGF-^198^AuNPs in managing the primary tumor location, a critical step in converting active disease to static disease and thus stopping metastases.

In order to estimate the tumor and local tissue doses in MGF-^198^AuNPs for prostate cancer radiotherapy in human patients, we have recently performed Monte-Carlo N-Particle code calculations. The overall objective of this investigation was to estimate the dose distribution delivered by radioactive gold nanoparticles (^198^AuNPs or ^199^AuNPs) to the tumor inside the human prostate, as well as to the normal tissues surrounding the tumor using Monte-Carlo N-Particle code (MCNP-6.1.1 code) [57]. According to the MCNP re-sults,^198^AuNPs present realistic potential for use in treating prostate and other solid cancers and for imaging purposes. In summary, the preclinical therapeutic efficacy studies, and the detailed toxicity studies of MGF-^198^AuNPs, provide compelling evidence for the clinical translation of this nanotherapeutic agent for use in treating prostate and related solid tumors in human patients. Therefore, future studies will focus on pilot clinical trials of MGF-^198^AuNPs, in prostate tumor bearing patients, in order to seek approval from regulatory agencies (FDA) for the utility of this new nanomedicine agent in oncology.

### 5.8. Clinical Trials of MGF-AuNPs in Human Cancer Patients

As discussed above, the production of MGF-^198^AuNPs is inherently associated with the presence of non-radioactive analogue MGF-AuNPs (Figure 9). In fact, samples of MGF-^198^AuNPs contain about 95% of the non-radioactive gold atoms as MGF-AuNPs. The interaction of radioactive Au-198 atoms with the non-radioactive Au atoms could generate a highly potent cocktail of therapeutic doses from photoelectric, Auger electrons and *X* rays. It was therefore important to investigate the therapeutic effects of the non-radioactive MGF-AuNPs through a pilot clinical trials investigation in human patients. After fully establishing doses at which MGF-AuNPs caused no toxicity concerns in vivo, a pilot clinical trial of this nanomedicine agent in human patients with triple negative breast cancers was initiated. This pilot human clinical trial was approved by AYUSH—a government of India regulatory agency.

Detailed in vitro and in vivo investigations in breast tumor bearing mice established unequivocally that MGF-AuNPs are highly effective in controlling the growth of breast tumors in a dose dependent fashion [104]. These encouraging pre-clinical results prompted seeking permission from regulatory authorities (AYUSH—a government of India regulatory agency) for conducting clinical trials in human patients. Patients treated with a MGF-AuNP drug cocktail along with the “standard of care treatment” (Arm B) exhibited measurable clinical benefits compared to patients in the study (Arm A) who received only the standard of care treatment [104]. The bar chart in Figure 9 shows that treatment of patients with MGF-AuNPs produced measurable therapeutic efficacy in reducing the tumor burden by 15–20%, as compared to the patient set which received only the standard of care chemotherapy treatment. These results indicate the tremendous clinical benefits of using mixtures of MGF-AuNPs and MGF-^198^AuNPs in clinically translating phytochemicals functionalized gold nanoparticles for treating various types of human cancers (Figure 9). Therefore, our extensive pre-clinical and clinical results, taken together, provide complete evidence that the mangiferin functionalized radioactive gold nanoparticles present realistic promise as a new nanoradiotherapeutic agent for treating prostate, breast and various types of solid tumors in human patients.

## 6. Conclusions

The CRP entitled ‘**Nanosized delivery systems for radiopharmaceuticals’** which was initiated in 2014 has resulted in major discoveries, and breakthroughs—all leading to the development of new chemical and radiation-based techniques for the development of a number of metallic and non-metallic nanoparticles. Here, we have attempted to summarize progress achieved in this CRP on the following fronts: (a) radionuclides for NPs radiolabeling; (b) design and development of NPs; (c) chemical-based synthesis of NPs—general aspects of production; (d) radiation-based synthesis of NPs; and (e) production, characterization, and pre-clinical/clinical investigations of radioactive MGF-^198^AuNPs. The radiolabeling protocols, tumor specific peptides and tumor avid small molecules developed for radiolabeling polymeric NPs, hydrogels and metallic NPs—have, individually and collectively, generated new science and a nanotechnology knowledge base for use in the design and development of next generation nanoradiopharmaceuticals.

## Figures and Tables

**Figure 1 pharmaceutics-14-01060-f001:**
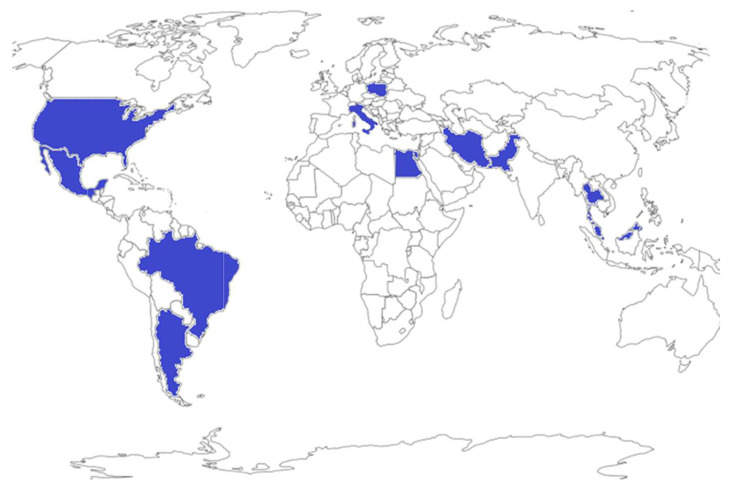
International distribution of the project participants from the member states.

**Figure 2 pharmaceutics-14-01060-f002:**
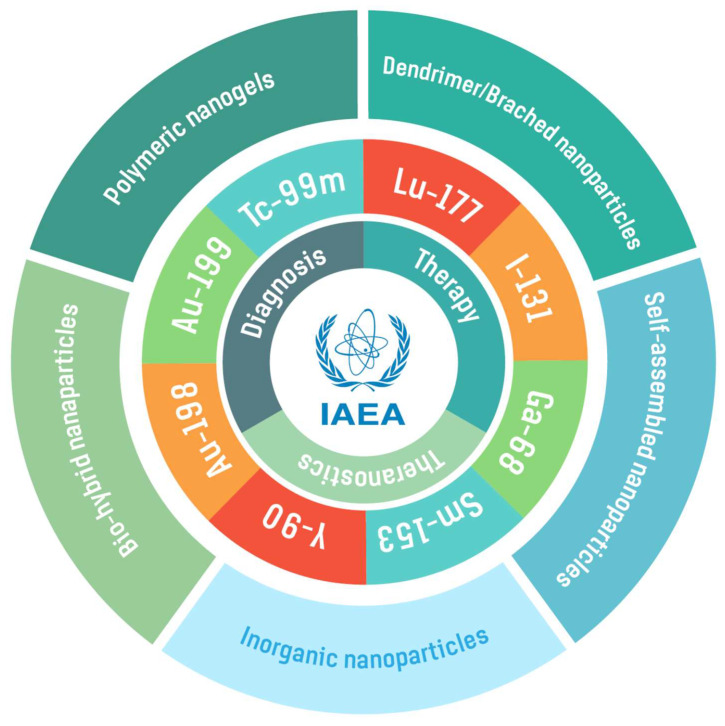
Overall focus of the review showing the development of a myriad of polymeric, organic, and inorganic radioactive NPs and varieties of diagnostic and therapeutic radioisotopes used in the architecture of nanoradiopharmaceuticals.

**Figure 3 pharmaceutics-14-01060-f003:**
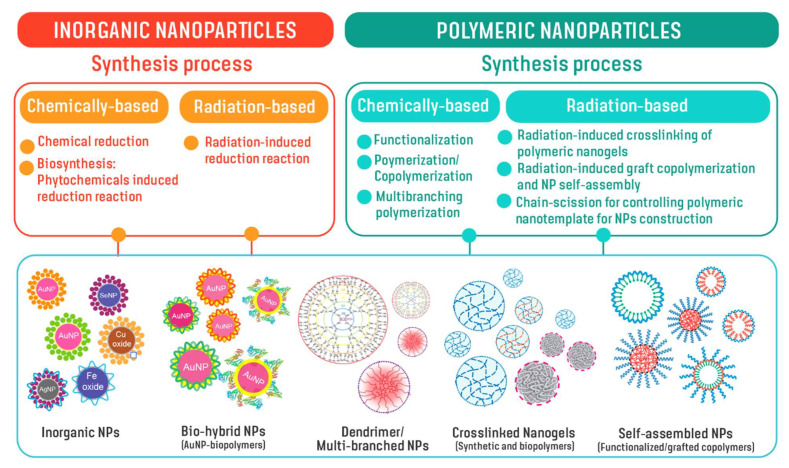
Structurally-controlled NPs developed through chemical- and radiation-based techniques for use in the creation of nanoradiopharmaceutical delivery approaches.

**Figure 4 pharmaceutics-14-01060-f004:**
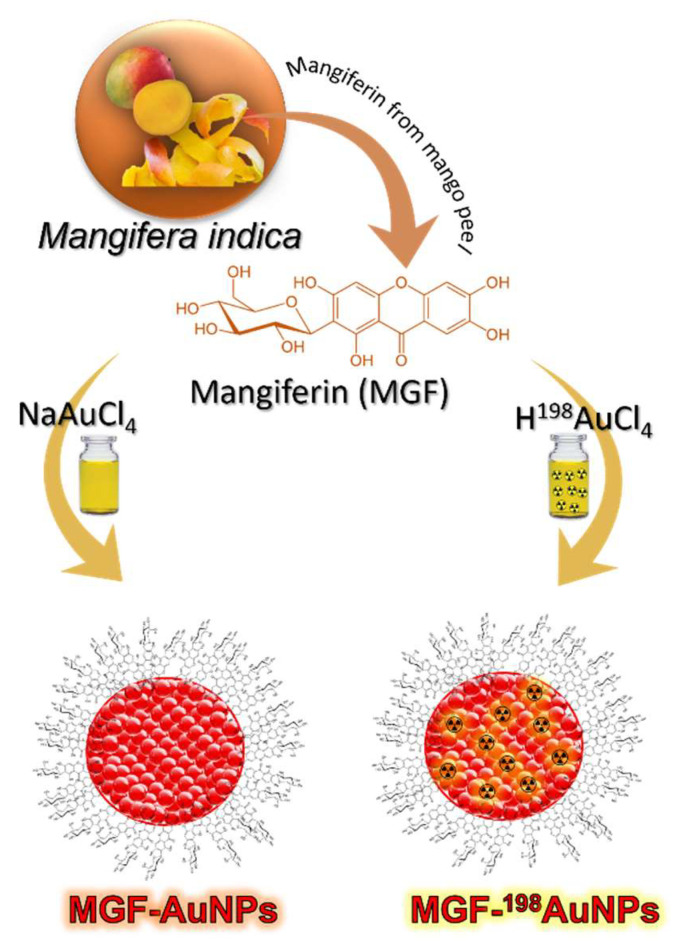
Synthesis and MGF corona architecture of MGF-AuNPs and MGF-^198^Au NPs.

**Figure 5 pharmaceutics-14-01060-f005:**
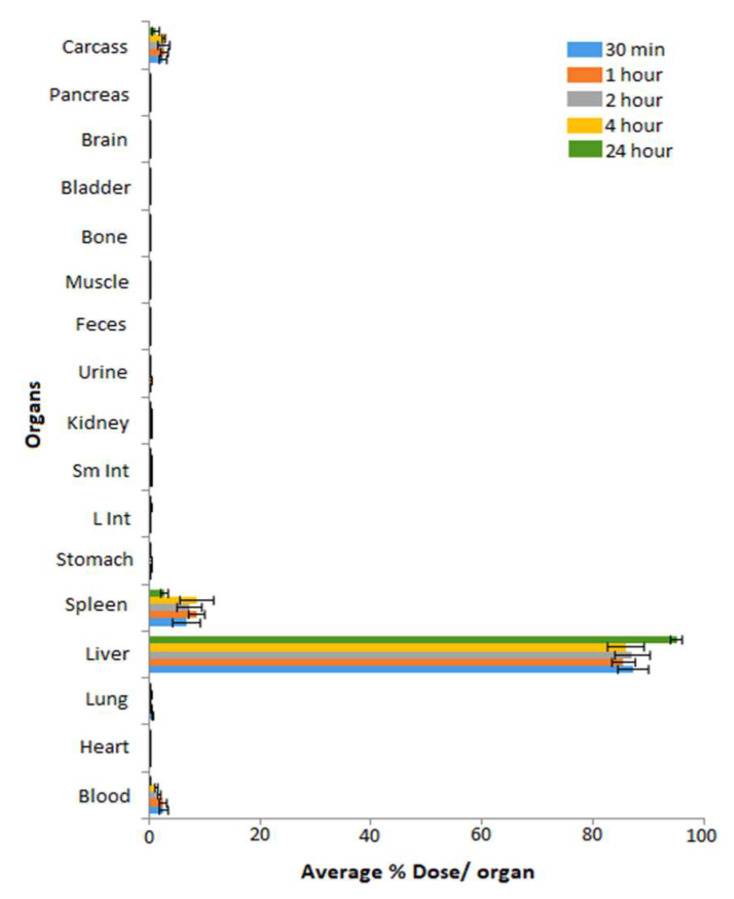
Biodistribution profile of MGF-^198^AuNPs in normal mice after intravenous administration of a single dose (296 kBq/100 μL) intravenously through a tail vein. Radioactivity was measured at 30 min, 1 h, 2 h, 4 h, and 24 h post-injection, and was calculated as the percentage of injected dose (%ID) per organ [38].

**Figure 6 pharmaceutics-14-01060-f006:**
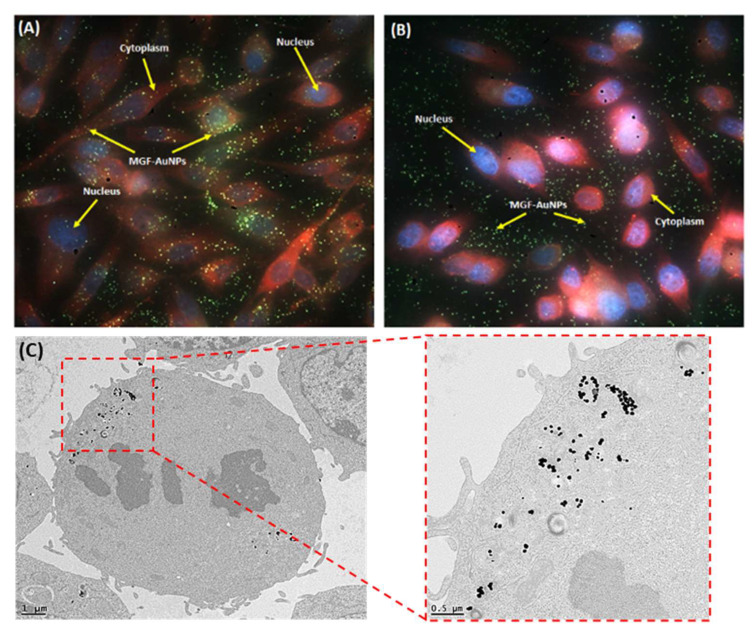
Dark field microscopic images (40×) investigating Lam 67 receptor avidity of mangiferin gold nanoparticles (MGF-AuNPs) towards prostate cancer cells. (**A**) PC-3 cells treated with MGF-AuNPs (8.2 μg/mL) and 2 hr incubation; (**B**) identical PC-3 cells treatment with MGF-AuNPs, following pre-treatment with laminin receptor antibody. Nuclei appear blue, cytoplasm red and NPs green in the images. (**C**) TEM images corroborate endocytosis of MGF-AuNPs in PC3-cells.

**Figure 7 pharmaceutics-14-01060-f007:**
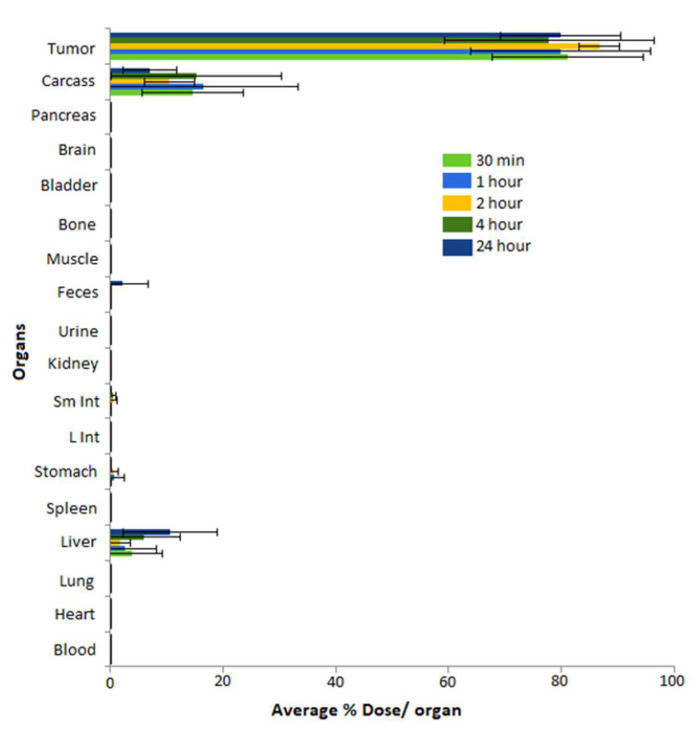
Tumor selectivity and retention of MGF-^198^AuNPs in tumors at 30 min, 1 h, 2 h, 4 h, and 24 h after direct injection of single dose of MGF-^198^AuNPs (4.0 µCi/30 µL) in prostate tumor. In this figure, radioactivity obtained from different organs was calculated as the percentage of injected dose (%ID) of each organ using SCID mice implanted with prostate tumor xenografts.

**Figure 8 pharmaceutics-14-01060-f008:**
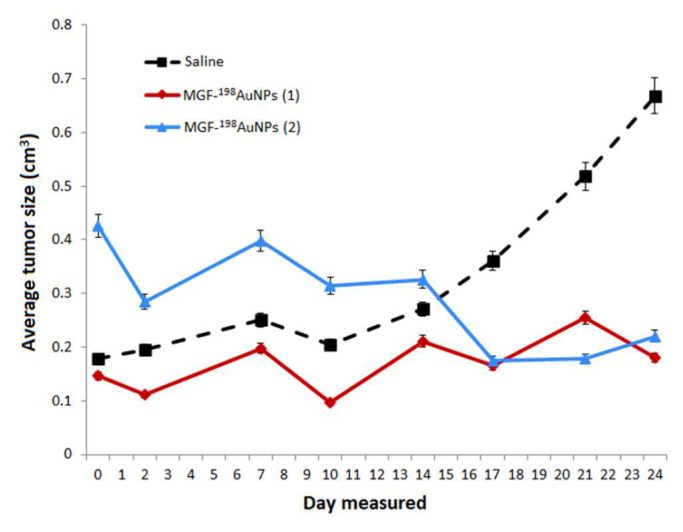
Therapeutic efficacy studies of MGF-^198^AuNPs after a single dose intratumoral administration in human prostate cancer bearing SCID mice (mean ± SD). By day 24, treated animal tumors were much smaller than the saline treated control group (*p* = 0.04). The therapeutic effect was maintained over a three-week period.

**Figure 9 pharmaceutics-14-01060-f009:**
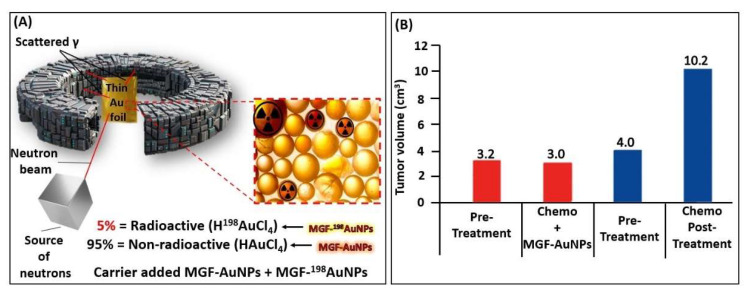
**(A) Production of carrier-added ^198^Au; (B) Mean tumor volumes from** clinical investigations of the non-radioactive MGF-AuNPs analogue in human triple negative breast cancer patients. The bar chart showing average tumor burden pre- and post-treatment [104].

**Table 2 pharmaceutics-14-01060-t002:** List of functionalized nanoparticles developed through chemical-based synthesis.

Categories of NPs	Name of NPs	Size (nm)	Information of NPs	Ref.
Inorganic	AgNP	21	AgNP prepared using ascorbic acid reducing agent and polyethylene glycol stabilizing agent for delivering ^131^I radioisotope.	[52]
	SeNP	43	SeNP prepared using sodium dithionate reducing agent and glutathione stabilizing agent for ^99m^Tc.	[49]
		23	SeNP prepared using sodium dithionate reducing agent and ascorbic acid as stabilizing agent for delivering ^99m^Tc radioisotope.	[50]
	AuNP	30–85	AuNP prepared using mangiferin as reducing and stabilizing agents for delivering ^198^Au radioisotope.	[42]
		20.3	AuNP prepared using Trisodium citrate as reducing and stabilizing agents for delivering ^99m^Tc radioisotope.	[44,47]
		50	AuNP prepared using gallic acid as reducing and stabilizing agent for delivering ^99m^Tc radioisotope.	[22,45,60]
	Fe oxide NP	24	Fe oxide NPs prepared by co-precipitation method and using polyethylene glycol as stabilizing agent for delivering ^99m^Tc radioisotope.	[51]
	Cu oxide NP	32.4	Cu oxide NPs prepared by biological synthesis using *Aspergillus flavus* for delivering ^99m^Tc.	[48]
Polymers	^99m^Tc-PAMAM-Tyr^3^-Octreotide ^99m^Tc -AuNP-Tyr^3^-Octreotide.	20 ^b^	Dendrimer-based or gold-based nanoradiopharmaceuticals for somatostatin receptors imaging on neuroendocrine tumors.	[17,46]
	^177^Lu-DenAuNP-folate-bombesin	18.60 ± 8.00 ^b^	^177^Lu-dendrimer (PAMAM-G4)-folate-bombesin with AuNPs in the dendritic cavity for targeted radiotherapy and the simultaneous detection of folate receptors (FRs) and gastrin-releasing peptide receptors (GRPRs) overexpressed in breast cancer cells.	[46,55,56]
	DOX-PLGA/γ-PGA-FA	597 ± 45.0 ^a^	Poly(L-γ-glutamic acid) (γ-PGA) conjugated to modified folic acid (FA) as a targeting ligand for specific doxorubicin delivery.	[61]
	PMAA(PTX)-RGD	17.5 ± 7.4 ^b^	Multimeric system of RGD-grafted PMMA- nanoparticles as a targeted drug-delivery system for paclitaxel.	[62]
	^177^Lu-PLGA(PTX)-BN	163.54 ± 33.25 ^a^	A targeted paclitaxel delivery system with concomitant radiotherapeutic effect for the treatment of GRPr-positive breast cancer.	[30]
	^177^Lu-DOTA-HA-PLGA(MTX)	167.6 ± 57.4 ^a^	Multifunctional chemo/radiotherapy agent based on PLGA, modified with hyaluronic acid and DOTA as a chelating agent for radiosynovectomy and specific targeted anti-rheumatic therapy.	[54]
	^177^Lu-DOTA-DN(PTX)-BN	16.37 ^a^	Bombesin targeted polymeric NPs designed to produce radiotherapy and chemotherapy towards GRP receptors.	[29,55]
	^177^Lu-DN(C19)-CXCR4	67.0 ± 23.17 ^a^	Nanoradiopharmaceutical dendrimer-based for combinatorial therapy (targeted chemotherapy and radiotherapy) in pancreatic cancer cell lines overexpressing the CXCR4 receptor.	[31]

^a^ Hydrodynamic size, ^b^ Size determined by TEM, ^c^ Size determined by AFM.

## Data Availability

Extensive literaure references have been cited in the review article.

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
