# Peer review of "IAEA Contribution to Nanosized Targeted Radiopharmaceuticals for Drug Delivery"

_pharmaceutics, 2022, doi:10.3390/pharmaceutics14051060_

Round 1
Reviewer 1 Report
The paper reviews the main achievements in the research of Nanosized Targeted Radiopharmaceuticals for Drug Delivery performed under the auspices of IAEA.
The description of the importance of nanosized radiopharmaceuticals for cancer therapy and of the different type of nanoparticles investigated is interesting and informative.
However, point 5 includes the detailed description of the Production, characterization, and pre-clinical/Clinical investigations of radioactive MGF-198AuNPs including experimental details that are extensively detailed in reference 38 of this paper.
The authors do not justify the selection of this specific NPs as example.
In my opinion detailed experimental data contained in another published paper should be avoided and this part should include a more general description of the aim and results obtained with this specific NPs. Furthermore, perhaps it would be better to include more examples and not only one for a detailed discussion and exemplification of the importance of the work performed with the support of IAEA.
Some minor details:
Table 1 is quite confusing because of the many abbreviations used in the column of type of nanoparticles. Perhaps it would be better to eliminate this column since this Table is intended to ilustrate the radionuclides used for labelling various nanoradiopharmaceuticals and the specific types of nanoparticles are included in Table 2 and 3.
Figures 5 and 6 are not very clear.
Also the biodistribution results in normal animals are not discussed.
In the description of biodistribution results in tumours the authors claim that the uptake in tumour increases at 2 hours. However taking into account the errors there is no significant difference between uptake and 30 min and 2 hours ( 80.98±13.39% at 30 min and 86.68±3.58% at 2 h).
Finally, when describing the pilot human clinical trial approved by regulatory agencies the authors do not specify which were the regulatory agencies that approved the study and in relation to the clinical trial they mention AYUSH without specifying if this is a regulatory agency and from which country.
Author Response
We attached our responses as a separate file

Reviewer 2 Report
The authors should be commended for their review about nanosized targeted radiopharmaceuticals for drug delivery, which could potentially frame the future of radiomolecular theranostics.
Author Response
We sincerely appreciate the comments of this examiner praising our work reported in this review.
Reviewer 3 Report
The review covers a topic of high scientific interest and describes a number of interesting results and hypotheses. Nevertheless, it is not always easy to understand what it aims to describe. This difficulty probably derives from a difficult language and from a series of typos that prevent an adequate comprehension. As an example, the data reported in Figure 9 are difficult to understand with this limitation further aggravated by the fact that the same Figure 9 is not quoted in the text. Finally, several concepts raised by the paper are strictly related to the nuclear physics environment (e.g. Auger electrons, please Auger not auger). Many of these terms might need an introduction considering the readership of the journal.
Author Response
See attached file for our responses to this reviewers comments
